# Physical Activity Behaviors of Children Who Register for the Universal, State-Wide Active Kids Voucher: Who Did the Voucher Program Reach?

**DOI:** 10.3390/ijerph17165691

**Published:** 2020-08-06

**Authors:** Bridget C. Foley, Katherine B. Owen, William Bellew, Luke Wolfenden, Kathryn Reilly, Adrian E. Bauman, Lindsey J. Reece

**Affiliations:** 1SPRINTER, Prevention Research Collaboration, Sydney School of Public Health, Faculty of Medicine and Health, The University of Sydney, D17 Charles Perkins Centre, Level 6, the Hub, Camperdown, NSW 2006, Australia; katherine.owen@sydney.edu.au (K.B.O.); william.bellew@sydney.edu.au (W.B.); adrian.bauman@sydney.edu.au (A.E.B.); lindsey.reece@sydney.edu.au (L.J.R.); 2The School of Medicine and Public Health, The University of Newcastle, Callaghan, NSW 2308, Australia; luke.wolfenden@health.nsw.gov.au (L.W.); kathryn.reilly@health.nsw.gov.au (K.R.)

**Keywords:** financial incentive, policy, organized sport

## Abstract

Active Kids is a government-led, universal voucher program that aims to reduce the cost of participation in structured physical activity for all school-enrolled children in New South Wales (NSW), Australia. As part of the Active Kids program evaluation, this cross-sectional study examined the Active Kids’ program’s reach to children in NSW and their physical activity behaviors, before voucher use. Demographic registration data from all children (4.5–18 years old) who registered for an Active Kids voucher in 2018 (*n* = 671,375) were compared with Census data. Binary and multinomial regression models assessed which correlates were associated with meeting physical activity guidelines and participation in the sessions of structured physical activity. The Active Kids program attracted more than half (53%) of all eligible children in NSW. Children who spoke a primary language other than English at home, were aged 15–18 years old, lived in the most disadvantaged areas, and girls, were less likely to register. Of the registered children, 70% had attended structured physical activity sessions at least once a week during the previous 12 months, whilst 19% achieved physical activity guidelines. Active Kids achieved substantial population reach and has the potential to improve children’s physical activity behaviors.

## 1. Introduction

Physical activity is associated with many health and wellbeing benefits [1,2]. For those aged 5–17 years, 60 min of moderate to vigorous intensity each day is recommended [3,4]. Guthold et al. estimated that most (81%) Australian children failed to meet physical activity guidelines [5]. Children may accumulate physical activity in various ways through activities of daily living, including school, active travel, and recreational physical activities, including play and structured sessions (e.g., sport, active recreation) [6]. Structured physical activities involve moderate to vigorous intensity activity and develop a range of physical, psychological, social, and cognitive skills required to lead a healthy life [1,2]. It is estimated that 47% of Australian children participate in less than one session of structured physical activity outside of school each week [7]. As age increases, participation in structured physical activity decreases, often remaining low throughout adulthood. A variety of people, communities, organizations, policies, and the wider environmental factors create barriers to structured physical activity participation in Australia. There is a need to increase structured physical activity participation during childhood and facilitate lifelong participation behaviors [8].

Effective government policies and interventions that aim to overcome systemic barriers to positive health behaviors are vital but, take time to implement and scale-up [9]. Although government-led actions have the potential to alter environments or social norms, these approaches are often met with resistance from the private sector or community organizations. For example, government taxation on tobacco products is proven to change purchasing behaviors and reduce smoking rates, yet some nations are taking a long time to implement this effective approach. Resistance to public policy interventions occurs when a profitable organization or group is likely to be negatively impacted by the positive public health intervention, or when there is limited community support. Such resistance can slow implementation, reduce the potential reach and mediate the expected effects [9]. Where health promoting government-led interventions are passed and implemented, comprehensive evaluations are important to continue advancing public health [10]. It is widely accepted that understanding reach is a critical part of evaluation, among other factors described by Glasgow et al. [11]. Reach should measure the number and characteristics of people who engage and whether they are representative of the target population faced by the barriers which the policy or the intervention aims to overcome [11].

In the context of children’s structured physical activity participation, financial incentive interventions that aim to reduce the cost of structured physical activity participation have gained political interest [12]. Cost is a major barrier that is stopping children from starting or continuing to participate in structured physical activities [13]. A previous example of a financial incentive for children’s structured physical activity implemented and evaluated, at scale, is the Canadian Fitness Tax Credit. Findings showed limited reach to children residing in disadvantaged areas [14]. The tax credit reached 12.3% of the eligible population in the first year, unfortunately limiting reach to those who would benefit most in disadvantaged areas [14].

In Australia, the New South Wales (NSW) Government, invested in a universal financial incentive voucher program entitled ‘Active Kids’ [15,16]. The Active Kids voucher was offered to all school-enrolled children in NSW (4.5–18 years old), commencing 31 January 2018 [17]. Each voucher provides AUD$100 towards the cost of an ≥8-week membership or registration fee, with approved structured activity providers. Structured physical activities include opportunities delivered though an organization, which involve physical exertion, skill and/or hand-eye coordination as the primary focus of the activity [8]; but elements of competition are not essential. These might be undertaken as team or individual pursuits, such as sport participation (e.g., Football, Swimming, Athletics, Tennis) or active recreation (e.g., Dance, Martial Arts, bush survival skills, etc.). A complex pragmatic evaluation was integrated within the design of the Active Kids program from the outset [16]. This study examined the Active Kids registration data to understand the program reach. Physical activity behaviors of registered children in NSW were examined, with consideration also given to the correlates of physical activity. The subsequent implementation and efficacy of the Active Kids voucher program are reported separately.

## 2. Materials and Methods

This study adopted a cross-sectional study design. All school-enrolled children (4.5–18 years old, N = 1,263,454) who lived in NSW and held a valid Australian universal healthcare number [18], were eligible for an Active Kids voucher between 31 January and 31 December 2018. Parents or carers registered children in the Active Kids program through a bespoke online government platform, which included standardized demographic data, physical activity, and health indicators. All registration data were extracted from the NSW Office of Sport registration database. The University of Sydney Human Research Ethics committee approved all ethical aspects of this study (Reference number: 2017/946).

Standardized demographic questions in the Active Kids registration form were sourced from government surveillance tools to ensure comparability and validity [19,20]. Demographic characteristics collected included child age, sex, primary language spoken at home, Aboriginal identity, disability status, socioeconomic status (SES), and remoteness. Disability status included physical, sensory, intellectual, psychiatric, or other health-related disabilities. SES was determined using postcode of residence and categorized using the Socio-Economic Index for Area, specifically the Index of Relative Socio-Economic Disadvantage [21], which ranks areas in Australia according to relative socioeconomic disadvantage. Remoteness was assessed using postcode of residence and categorized using the Accessibility and Remoteness Index of Australia (ARIA+). ARIA+ groups areas on the basis of relative access to services, into major city, inner regional, outer regional, or remote [22].

Achievement of physical activity guidelines was assessed using a single item question reported by the parent or the carer [23]. The item asked, “In a typical week, how many days was the child physically active for at least 60 min?” Response options were days between ‘0–7’ days or ‘not sure’. There is evidence that this is a valid and reliable self-report measure of physical activity in adolescents [23]. Children were classified as meeting physical activity guidelines if ‘7 days’ was selected [3,4,23].

Participation in sessions of structured physical activity was measured using a single item reported by the parent or carer [20]. The item asked, “Approximately, how many organized sessions of sport or physical activities has the child participated in, outside of school hours, during the last 12 months?” The parent or carer had the option to respond by entering the number of times in the last 12 months, number of times per month, or number of times per week. Children were classified into the following five categories by dividing the number of annual session by 52 for a weekly average number of sessions—’no participation’ (0 sessions/year), ‘at least once a month’ (<52 session/year), ‘at least once a week’ (52–103 session/year), ‘at least twice a week’ (104–207 sessions/year), ‘at least four times a week’ (≥208 sessions/year).

Parent/carer reported that the child height and weight were non-mandatory fields in the registration form. Height and weight were used to calculate Body Mass Index (BMI) for each child, which was categorized as thin, healthy weight, overweight, or obese, using the International Obesity Taskforce definitions [24].

Frequencies and proportions for demographic characteristics were calculated for all children in NSW and those who registered in the Active Kids program. Characteristics of eligible children were compared against the National 2016 census conducted by the Australian Bureau of Statistics [25]. Due to the large sample size, all associations between covariates and physical activity were significant. Therefore, proportional reporting ratios (PRR) were then calculated to better characterize the magnitude of differences between all eligible children and those registered in the program [26].

Binary logistic regression models were conducted to determine the demographic characteristics that were associated with meeting the physical activity guidelines, and the multinomial regression models to examine which demographic characteristics were associated with participation in sessions of structured physical activity. Analyses were performed in SAS Enterprise Guide 9.4 (SAS Institute, Cary, NC, USA).

## 3. Results

Between 31 January and 31 December 2018, 671,705 (53%) of all eligible school-enrolled children in NSW registered for an Active Kids voucher (Table 1). Active Kids achieved greater reach amongst young children (4–11 years old), children who identify as Aboriginal/Torres Strait Islander, who speak English at home, and who identified as having a disability. Boys registered in similar proportions to the total population while fewer eligible girls registered for a voucher. Children who spoke a primary language other than English at home, were aged 15–18 years old, lived in socio-economically disadvantaged areas, and girls, were less likely to be registered for an Active Kids voucher. Similar proportions of children from major cities, inner regional, and outer regional/remote areas registered in the Active Kids program. Census data for BMI was not available, however, 306,450 (46%) participants voluntarily chose to provide height and weight data.

### Correlates of Physical Activity and Structured Physical Activity Participation

Of the children who registered for an Active Kids voucher, 19.3% (*n* = 129,292) achieved health enhancing physical activity guidelines (Table 2). Most children (92%, *n* = 618,733) reported having participated in structured physical activity sessions outside-of-school during the past 12 months; 15.1% participated in structured physical activity sessions at least four times a week, 22.0% participated at least twice a week, 32.5% participated at least once a week, and 22.6% at least once a month (Table 2). Children who participated in more structured physical activity sessions had greater odds of meeting physical activity guidelines (Table 2).

Children who live in regional and remote areas and children who live in high socio-economic areas had higher odds of meeting physical activity guidelines and higher odds of participating in structured physical activity sessions, compared to children who live in a major city or low socio-economic areas. Higher odds of meeting physical activity guidelines but lower odds of participating in structured physical activities outside of school were observed among Aboriginal/Torres Strait Islander children, children categorized as ‘Thin’, and younger children compared to non-Aboriginal/Torres Strait Islander children, children categorized as ‘Healthy weight’, and children over nine years old. As the child’s age increased, their odds of participating in structured physical activities increased, whilst the odds of meeting physical activity guidelines reduced in older children. Lower odds of meeting physical activity guidelines and lower odds of participating in structured physical activities were observed among girls, children who speak a language other than English at home, children with a disability, and children categorized as ‘obese’ compared to boys, children who only speak English at home, children free from disability, and those who are not obese (Table 2).

## 4. Discussion

This is the first study to demonstrate the population reach of a government-led, universal children’s structured physical activity voucher intervention, delivered at scale. The Active Kids voucher program reached more than half of the eligible NSW state population, over 600,000 children in 2018. An important component of the complex pragmatic evaluation of the Active Kids program [16], this substantial reach in the program’s first year indicated the intervention’s ability to engage significant numbers of children and families. Furthermore, the characteristics of the population registered was largely representative of the NSW population. The significant reach was one indicator of success for this program and had the potential to change societal norms for school-aged children around physical activity and lifestyle behaviors.

Although the Active Kids voucher was universally available, children living in socioeconomically disadvantaged areas, who speak a language other than English at home, who are over the age of 15 years old, and girls, were underrepresented. In Australia, these underrepresented groups are also known to have lower structured physical activity participation rates [7]. These findings of the program reach were improved compared to the population-wide Children’s Fitness Tax Credit (CFTC), which reported reaching 20% of children in their highest income category, whilst reach in their low-income categories were less than 1% [14]. To ensure that existing inequities in physical activity behaviors were not widened, complementary actions that reduce barriers to structured participation should be addressed for underrepresented groups, such as increasing opportunities for culturally appropriate activities, targeted mass media campaigns to promote the voucher to these groups, and improved transport infrastructure to increase access to structured physical activity opportunities [8].

A large proportion of children who registered for a voucher had participated in a structured physical activity session in the previous 12 months, before the Active Kids voucher was available. Children who were most committed to sport and lived in high socioeconomic areas were most effectively reached by the Active Kids program. Their previous engagement in the sector might indicate that the voucher encouraged participants to return to structured sessions, or that the voucher availability did not change their behavior. Owen et al. (2020), found that children living in low socioeconomic areas were less likely to have heard of the Active Kids program and to have registered for an Active Kids voucher [27]. Again, in Canada, the CFTC has limited awareness and uptake among families in disadvantaged areas [14]. Both the CFTC and Active Kids interventions took an equality-based approach in their program design. In Canada, the financial incentive value was an ‘equal proportion of the amount spent on registration for all’, which meant that “the rich got richer” and the tax credit had little value for low-income families [14]. The Active Kids program provided an ‘equal value to all participants, independent of amount spent on the child’s registration’ [15]. The Active Kids program demonstrated much higher initial population reach to disadvantaged children (38%) than CFTC, likely due to a greater proportion of disadvantaged children’s expenses being supported by the $100 Active Kids voucher [12]. Differences in program design, stakeholder involvement, and implementation, also contribute to the different reach between programs.

A financial incentive is, however, only one part of a multi-component approach that is required to address population inactivity [8]. A recent systematic review highlights that cost, lack of time, peer relationships, and access to local opportunities to participate in structured physical activities are barriers that need to be addressed [13]. Additional components are required to achieve equitable program reach in socioeconomically disadvantaged areas, families who speak a language other than English at home and older children. These sub-groups are known to be less active than their counterparts and have the greatest potential to benefit from using a voucher [28,29]. Small-scale financial incentive studies propose co-design of eligible activities with children in low socioeconomic circumstances might increase their use of the voucher [30,31]. Intervention components that are delivered in partnership with stakeholders beyond the physical activity and sport sectors, such as mass media and communications campaigns targeted towards culturally and linguistically diverse families or adolescents, might further increase the intervention reach in the subsequent years of delivery of Active Kids [32].

Evaluation of large-scale policies and interventions should always be undertaken and used to monitor inequalities, informing delivery in real time [14,16]. The Active Kids registration dataset enabled daily monitoring of program reach during 2018. The government then used this data to create infographics to share with local stakeholders, including in areas with proportionately low registrations for the population. Locally relevant data on children’s registrations in the Active Kids program engaged stakeholders, enabling local strategies to promote Active Kids, which likely contributed to the high population reach at the end of the program’s first year. When interventions were scaled up, ongoing modifications and additional components were recommended to strengthen the intervention within a particular context [33]. The approach taken by the NSW Government for the Active Kids program, to integrate an independent evaluation within the program protocol, is an exemplar of a best practice evaluation approach, enabling evidence-based modifications and additional components during implementation

This study showed that children who participate regularly in structured activities have greater odds of achieving the recommended levels of physical activity. This result aligns with public opinion of Australia being a sporting nation [34], demonstrating the important role that structured physical activities can play in assisting some children to achieve the recommended physical activity guidelines, especially if they participate regularly. Longitudinal research provides further evidence that participation in structured sports programs, increases the odds of physical activity participation later in life [6]. The community sport and recreation sector should be encouraged by these findings to broaden their membership reach and encourage new children to participate in structured activities. Children who participate regularly in structured physical activities might still exhibit poor physical literacy [35,36]. In addition to increasing participation in their sessions, structured physical activity programs should focus on retaining participants and progressing them along the physical literacy continuum, encouraging lifelong participation.

Self-report physical activity participation is frequently monitored in relation to physical activity guidelines for moderate to vigorous physical activity. Our study found that 19% of all children registering for an Active Kids voucher met the physical activity guidelines, which is similar to 11 National and State data sources across Australia that estimated 15–41% of 5–17-year old children in Australia achieve physical activity guidelines [35]. The correlates observed in this study were similar to previous research, identifying greater inactivity among older adolescents, girls, children who speak a language other than English at home, children with a disability, living in metropolitan areas, or socioeconomically disadvantaged areas [36]. This low proportion of the population meeting physical activity guidelines was concerning and further justified the need for Active Kids, as well as additional public health interventions to increase children’s physical activity levels. The integration of both physical activity and sport participation measurement was a strength of this study.

Another strength of this study was the representative population sample that would provide policy makers, academics, and practitioners, with robust data to make judgments on the implementation and effectiveness of Active Kids, as well as understanding more about physical activity behaviors of the children engaged in the program. Recognizing that the Active Kids data set cannot be used as a surveillance tool, considering the program bias, it did provide rich data on the physical activity behaviors of children in NSW complementing other population data sets in Australia such as AusPlay [7]. The limitations of this study were the self-report nature of the data collected and the volunteer bias in children and parents who expressed interest in the Active Kids program. The proportion of children who reported participating in structured physical activity sessions in this study was higher than population estimates from AusPlay [7]. All outcomes reported for children and adolescents were completed by a parent or carer by proxy and were based on self-report data, making the data prone to social desirability bias and recall bias [37].

Population data available within Australia used significantly smaller samples; the National and State population health surveys included data from 21,300 people (4273 in NSW) and 12,000 people, respectively [19,38]; National sport sector survey, AusPlay, had an annual target sample size of 20,000 adults, and 3600 children (5922 adults and 1175 children in NSW) [20]. These ongoing population measures are important but might have insufficient sample-sizes for subgroup analysis. Subgroups that often lack statistical power in population surveillance of physical activity behaviors include Aboriginal/Torres Strait Islander children, culturally and linguistically diverse communities, children with a disability, and children in regional and remote areas [39,40]. This Active Kids dataset enabled some of these gaps to be filled with the methodological limitations and for bias to be acknowledged. This dataset provided relevant information on the initial reach of the Active Kids program and will help assess participation and maintenance of physical activity in subsequent years [16].

Contrary to known reductions in structured physical activity participation with increasing age [41], in our sample participation was high among other age groups, with highest odds of participation at age 12–14 years and participation remained high among 15–18-year-olds. Declining rates of sport participation during adolescence are well established in the literature, therefore, the interpretation of this finding should consider the smaller proportion of 15–18-year-olds represented in the study sample, those likely to be sport-focused, reinforcing the bias acknowledged above. Older children, therefore, who did not register for a voucher, were a priority population, based on previous research highlighting the risk of drop-out [5]. Another contradictory finding to the existing literature was from the 36,129 Aboriginal Australian and Torres Strait islander children in this sample. This dataset provides the largest sample of Aboriginal and Torres Strait Islander children’s physical activity behaviors and suggests that most Aboriginal and Torres Strait Islander children in NSW do not regularly participate in structured physical activities. This requires further investigation, as previous research described the cultural significance of sport and physical activity among Aboriginal and Torres Strait Islander communities [42]. The size and scale of Active Kids with its myriad of delivery partners also presented a unique opportunity to drive towards a consistent measurement system for physical activity [43].

## 5. Conclusions

The universal nature of the Active Kids voucher has the potential to overcome systemic barriers to structured physical activity participation for children. The first year of the Active Kids program achieved substantial reach to over 53% of the NSW population of school-enrolled children. To enhance reach to under-represented groups, the state government should strengthen collaborations and strategic partnerships with organizations in sport and other sectors, including education, justice, multicultural, and community services. The integration of standardized questions within the Active Kids registration platform provides unique insights into physical activity behaviors of more than half the NSW child population. Inequities in physical activity participation identified at registration have practical implications for how the wider-community and Active Kids providers engage disadvantaged children in structured physical activity programs. Additional efforts should be made to ensure less active children, who register for a voucher, actually use their Active Kids voucher to participate in eligible programs. Further evaluation throughout the duration of the program implementation should examine the effect of the Active Kids voucher on children’s physical activity participation and additional understanding of the mediating factors that influence children’s participation in structured physical activity.

## Figures and Tables

**Table 1 ijerph-17-05691-t001:** Reach of the Active Kids program in New South Wales.

Demographic Characteristics	All Eligible Children in New South Wales N = 1,263,454		Children who Registered in the Active Kids in 2018 N = 671,375 (53.2%)		Difference Between Groups
	N	%	N	%	PRR
**Age category**					
4–8 years	378,787	30.0	269,457	71.1	1.3
9–11 years	274,038	21.7	185,931	67.8	1.3
12–14 years	258,828	20.5	138,063	53.3	1.0
15–18 years	351,801	27.8	77,924	22.2	0.4
Sex *					
Boys	648,759	51.3	361,852	55.8	1.0
Girls	614,695	48.7	308,543	50.2	0.9
Primary language spoken at home					
English	953,924	75.5	621,535	65.2	1.2
Other	309,530	24.5	50,140	16.2	0.3
Aboriginal identity					
Aboriginal/Torres Strait Islander	59,554	4.7	36,129	60.7	1.1
Non-Aboriginal/Torres Strait Islander	1,203,900	95.3	626,688	52.1	1.0
Prefer not to say			8558		
Disability					
Yes	31,705	2.5	17,715	55.9	1.1
No	1,169,846	92.6	644,658	55.1	1.0
Prefer not to say			9002		
Socio-economic status ^					
1st quartile (Most disadvantaged)	263,911	20.9	99,583	37.7	0.7
2nd quartile	290,625	23.0	140,302	48.3	0.9
3rd quartile	334,919	26.5	158,783	47.4	0.9
4th quartile (Most advantaged)	373,455	29.6	200,566	53.7	1.0
Missing			72,141		
Location ^					
Major City	935,525	74.0	440,793	47.1	0.9
Inner Regional	257,961	20.4	126,594	49.1	0.9
Outer Regional and remote	69,943	5.5	32,622	46.6	0.9
Missing			71,366		
Body Mass Index Category					
Thin	NA		35,357		
Healthy weight	NA		195,166		
Overweight	NA		52,675		
Obesity	NA		23,252		
Missing	1,263,454	100.0	365,255	28.9	0.5

* Some Active Kids participants did not report sex (<0.2%). ^ Some postcodes were missing or invalid (11% for socioeconomic status) (11% for geographic location). NA: Data not available from the national Census [25].

**Table 2 ijerph-17-05691-t002:** Odds ratios of children at registration meeting physical activity guidelines and participating in structured physical activities.

Characteristic	Physical Activity	Structured Physical Activity Participation			
	Met Guidelines Odds Ratio (95% CIs)	At least Once a Month Odds Ratio (95% CIs)	At least Once a Week Odds Ratio (95% CIs)	At least Twice a Week Odds Ratio (95% CIs)	At least Four Times a Week Odds Ratio (95% CIs)
Total N (%)	129,292 (19.3%)	151,758 (22.6%)	217,963 (32.5%)	147,696 (22.0%)	101,316 (15.1%)
Physical Activity Guidelines					
Met guidelines	**	1.05 (0.99, 1.11)	1.26 (1.20, 1.34)	1.76 (1.67, 1.86)	3.88 (3.67, 4.10)
Age category					
4–8	Ref	Ref	Ref	Ref	Ref
9–11	0.69 (0.68, 0.71)	1.21 (1.16, 1.27)	1.42 (1.36, 1.49)	1.88 (1.79, 1.97)	3.15 (3.00, 3.30)
12–14	0.47 (0.47, 0.48)	1.24 (1.17, 1.31)	1.44 (1.37, 1.52)	2.01 (1.90, 2.12)	4.02 (3.81, 4.25)
15–18	0.39 (0.38, 0.40)	1.19 (1.11, 1.28)	1.38 (1.29, 1.47)	1.89 (1.77, 2.03)	4.01 (3.74, 4.30)
Sex					
Boys	Ref	Ref	Ref	Ref	Ref
Girls	0.61 (0.60, 0.61)	0.88 (0.85, 0.92)	0.83 (0.80, 0.86)	0.82 (0.79, 0.85)	0.79 (0.77, 0.83)
Primary language spoken at home					
English	Ref	Ref	Ref	Ref	Ref
Other	0.68 (0.67, 0.7)	0.60 (0.57, 0.63)	0.47 (0.44, 0.49)	0.35 (0.33, 0.37)	0.23 (0.22, 0.25)
Aboriginal identity					
Aboriginal/Torres Strait Islander	1.31 (1.28, 1.35)	1.03 (0.95, 1.10)	0.69 (0.64, 0.74)	0.66 (0.61, 0.71)	0.76 (0.70, 0.82)
Non-Aboriginal/Torres Strait Islander	Ref	Ref	Ref	Ref	Ref
Disability					
Yes	0.88 (0.84, 0.91)	0.59 (0.54, 0.64)	0.38 (0.35, 0.41)	0.28 (0.26, 0.3)	0.21 (0.19, 0.22)
No	Ref	Ref	Ref	Ref	Ref
Socio-economic status					
1st quartile (Most disadvantaged)	Ref	Ref	Ref	Ref	Ref
2nd quartile	1.09 (1.07, 1.11)	1.31 (1.24, 1.38)	1.63 (1.54, 1.71)	1.70 (1.61, 1.80)	1.70 (1.61, 1.80)
3rd quartile	1.08 (1.06, 1.10)	1.54 (1.47, 1.63)	2.17 (2.06, 2.28)	2.31 (2.19, 2.43)	2.22 (2.10, 2.34)
4th quartile (Most advantaged)	1.16 (1.13, 1.18)	2.14 (2.02, 2.27)	3.69 (3.49, 3.91)	4.27 (4.02, 4.52)	4.11 (3.87, 4.36)
Location					
Major Cities	Ref	Ref	Ref	Ref	Ref
Inner Regional	1.37 (1.34, 1.39)	1.32 (1.25, 1.39)	1.32 (1.26, 1.39)	1.30 (1.23, 1.36)	1.24 (1.18, 1.31)
Outer Regional and remote	1.55 (1.50, 1.59)	1.86 (1.69, 2.04)	1.80 (1.64, 1.97)	1.81 (1.65, 1.99)	1.70 (1.54, 1.87)
Body Mass Index category					
Thin	1.08 (1.05, 1.11)	0.87 (0.79, 0.96)	0.74 (0.68, 0.82)	0.70 (0.63, 0.77)	0.68 (0.62, 0.75)
Healthy weight	Ref	Ref	Ref	Ref	Ref
Overweight	0.78 (0.76, 0.80)	1.01 (0.93, 1.11)	0.89 (0.81, 0.97)	0.80 (0.73, 0.88)	0.70 (0.64, 0.77)
Obesity	0.67 (0.65, 0.70)	0.84 (0.76, 0.93)	0.58 (0.52, 0.64)	0.45 (0.4, 0.49)	0.36 (0.33, 0.41)

** The reference group for physical activity guidelines is ‘Did not achieve guidelines’, i.e., achieved less than 60 min of physical activity per day in the past 7 days. The reference group for structured physical activity participation is ‘zero sessions in past 12 months’.

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
