# Peer review of "Physical Activity Behaviors of Children Who Register for the Universal, State-Wide Active Kids Voucher: Who Did the Voucher Program Reach?"

_ijerph, 2020, doi:10.3390/ijerph17165691_

Round 1

Reviewer 1 Report

The article is interesting but I would like to make the following recommendations:
- I recommend that the title be rewritten, more suggestive and the elimination of the actual number
- In table 1, I think it is an error, if we add the number of children by sex is not 671,375 but 670,395, I recommend checking
- Under table 1 I recommend you to specify what NA means
- I recommend to specify clearly which are the physical activities included in this voucher.
I believe that in order for this article to be accepted in this journal, it should have provided complete data, the completion of this action to stimulate the practice of physical activities.

Reviewer 2 Report

I enjoyed reading this well-written paper in a public health area. The aim of this study was to  examine the Active Kids’ program’s reach to children in New South Wales and their physical activity behaviors before voucher use. I have some minor issues to consider by authors:

*Introduction

1. line 36 „(Physical literacy ref)” - should the references be cited here?

2.“Each voucher provides AUD$100 towards the cost of membership or registration fees with approved  structured activity providers.” - for what period of time membership was established?

*Materials and Methods

  1. Line 75: „All school-enrolled children (4.5-18 years old) who lived in NSW” – please, state here how many children lived in NSW in 2018.
  2. Line 93-96: please provide the adopted in the study definition of meeting the physical activity guidelines and quote the relevant reference.

*Results

  1. Line 130: I would delete a word “only” (the same from Abstract)
  2. Table 2. The first row of the table (header) is not clear to me. Whether the signature: “Met physical activity guidelines Odds ratio (95% CIs)” means that study participants met guidelines every day? (please look at my comment no 2 from previous section). Similarly the other headers  in this table, for example “At least once a month Odds ratio (95% CIs)”- does it mean “meeting physical activity guidelines at least one a month”?

Reviewer 3 Report

The manuscript entitled “Who registered for an Active Kids voucher? Analysis of 671,375 children’s physical activity behaviors” comprises a cross-sectional study that examines the Active Kids’ programs reach to children in NSW and their physical activity behaviors before voucher use. The readability of the manuscript is good and comprises a significant advance in the comprehension of the program reach in this population. Nevertheless, there are some minor issues that need to be addressed before publication can be considered:

Specific comments:

  • Page 1, line 6 – what “physical literacy ref” means?
  • Results sections – please provide the odds in the text for readers to better follow the table;
  • Page 4, lines 31-33 – I did not found this information in the table;
  • Page 4, line 39 – please provide reference categories in the text so that readers can easily follow;
  • In table 2, the categories of the subjects participating in structured physical activities do not match with the description in the material and methods section: "The parent or career had the option to respond by entering the number of times in the last 12 months, number of times per month, or number of times per week.” How authors’ converted answers in these categories? Please provide more detailed information.
  • In table 2, the last row “physical activity guidelines” should be allocated for the second row of the table. This is one of the most important results and they are highlighted first in the text.
  • Page 7, lines 160-163 – The intervention made in CANADA (Children’s Fitness Tax Credit (CFTC)) previously reported in the introduction section presented similar results? Or are results unavailable?
  • Page 7, line 165 – Please provide examples or suggestions of such putative actions used to reduce barriers;
  • Conclusions section - Please provide more information regarding the practical implications of the results;

Round 2

Reviewer 1 Report

no comments